# The mitochondrial proteomic changes of rat hippocampus induced by 28-day simulated microgravity

Guohua Ji[1]*, Hui Chang[2]*, Mingsi Yang[2]*, Hailong Chen[1], Tingmei Wang[1], Xu Liu[2], Ke Lv[1], Yinghui Li[1], Bo Song[2]*, Lina Qu[1]*

1 State Key Laboratory of Space Medicine Fundamentals and Application, China Astronaut Research and Training Center, Beijing, China, 2 Department of Pathology and Forensics, College of Basic Medical Sciences, Dalian Medical University, Dalian, China

☯ These authors contributed equally to this work.
* linaqu@263.net (LQ); bosong@dmu.edu.cn (BS)

## Abstract

A large number of aerospace practices have confirmed that the aerospace microgravity environment can lead to cognitive function decline. Mitochondria are the most important energy metabolism organelles, and some studies demonstrate that the areospace microgravity environment can cause mitochondrial dysfunction. However, the relationships between cognitive function decline and mitochondrial dysfunction in the microgravity environment have not been elucidated. In this study, we simulated the microgravity environment in the Sprague-Dawley (SD) rats by -30˚ tail suspension for 28 days. We then investigated the changes of mitochondrial morphology and proteomics in the hippocampus. The electron microscopy results showed that the 28-day tail suspension increased the mitochondria number and size of rat hippocampal neuronal soma. Using TMT-based proteomics analysis, we identified 163 differentially expressed proteins (DEPs) between tail suspension and control samples, and among them, 128 proteins were upregulated and 35 proteins were downregulated. Functional and network analyses of the DEPs indicated that several of mitochondrial metabolic processes including the tricarboxylic acid (TCA) cycle were altered by simulating microgravity (SM). We verified 3 upregulated proteins, aconitate hydratase (ACO2), dihydrolipoamide S-succinyltransferase (DLST), and citrate synthase (CS), in the TCA cycle process by western blotting and confirmed their differential expressions between tail suspension and control samples. Taken together, our results demonstrate that 28-day tail suspension can cause changes in the morphology and metabolic function of hippocampus mitochondria, which might represent a mechanism of cognitive disorder caused by aerospace microgravity.

**Data Availability Statement:** All relevant data are within the paper and its Supporting Information files.

**Funding:** This work was supported by The Foundation of State Key Laboratory of Space Medicine Fundamentals and Application, China Astronaut Research and Training Center (SMFA17A03, SMFA19B02, SMFA19K08, SMFA20A01), The National Natural Science Foundation of China (21635001, 31800707, 31800998). The funders had no role in the study design, data collection and analysis, decision to publish, and preparation of the manuscript.

**Competing interests:** The authors have declared that no competing interests exist.

## Introduction

Long-term space travel would adversely affect human physiology, and the most common detrimental effects include the visually impaired intracranial pressure syndrome, decreased bone density, muscle atrophy, brain functional and structural changes [1, 2]. Since spaceflight missions are rare, the ground based analogues have been developed to simulate the space environment. Animal tail suspension is a classical and useful approach for the microgravity study on Earth for a long time [3–5]. There have been several reports about the impacts of simulated microgravity (SM) on brain cognitive function. Lin et al. demonstrated that SM inhibits the proliferation of adult hippocampal neural stem cells in rats, which maybe the reason of detrimental effects of SM on learning and memory [6]. Nday et al. summarized the recently published articles and concluded that the effects of SM on brain include brain plasticity, brain neurotrophic factor (GDNF), apoptosis factors (Bcl-xL and Bax), 5-hydroxytryptamine and dopaminergic system, and dopaminergic gene expression. The neuropathological characteristics of animal SM model can be comparable to the effects of aging, anxiety and other neurological diseases [7]. Our previous research also found that tail suspension can damage the learning and memory ability of rats, and the molecules involved in glutamate excitotoxicity and several neurotransmitters (5-hydroxytryptamine, dopamine, γ-amino acid butyric acid and epinephrine) are downregulated [5, 8].

Mitochondria are essential for aerobic eukaryotes and are the most important energy supply organelles [9, 10]. Mitochondria participate in key central metabolic pathways, and are fully integrated into intracellular signaling networks that regulate multiple cellular functions, including ATP production, regulation of excitotoxicity, intracellular $Ca^{2+}$ homeostasis, production of reactive oxygen species, release of cytochrome c and induction of cell apoptosis [11, 12]. The human mitochondrial genome (mtDNA) contains only 37 genes that encode 13 proteins [13]. The remaining thousands of mitochondrial proteins are encoded by the nuclear genome. Therefore, compared with the mtDNA, the mitochondrial proteome can provide a more comprehensive perspective for understanding mitochondrial functions. Multi-omics analysis of the hundreds of data from spaceflight for astronauts and rodents revealed that the mitochondrial processes as well as innate immunity, chronic inflammation, cell cycle, circadian rhythm, and olfactory functions are the most significant enrichment processes, and mitochondrial stress is the central biological hub of the impact of spaceflight on human beings [14]. However, the effect of SM on the hippocampus mitochondrial proteome has not been explored.

The mitochondrial dysfunction has also been demonstrated to participate in several diseases such as Alzheimer's disease [15], cancers [16], cardiovascular diseases [17], Parkinson disease [18], traumatic brain injury and epilepsy [19, 20] etc. Therefore, the study on the mechanism of mitochondrial dysfunction would be benefit for astronauts as well as the health of people on Earth. Herein, we examined the changes in the rat hippocampus mitochondrial proteome caused by the 28-day tail suspension, and performed Gene Ontology (GO) classification, pathway enrichment and protein-protein interaction analysis on the differentially expressed proteins (DEPs). Our research can provide inspiration for understanding the mitochondrial-related molecular mechanism of cognitive function decline in microgravity.

## Materials and methods

### Animals and SM model construction

Eight weeks male Sprague-Dawley (SD) rats of SPF grade were purchased from Beijing Vital River Laboratory Animal Technology Co. Ltd., China. All rats were kept in separate cages and

placed in a temperature controlled environment. The light dark cycle was 12/12 hours, and they had free access to food and water. All experimental procedures were approved by the Animal Care and Use Committee of China Astronaut Research and Training Center. After 2 weeks of adaptive feeding, the rats were randomly divided into two groups, Control group (C) and SM group (T), and each group contained 24 rats. The method of tail suspension to SM effect was described previously [21], and the Control group was raised in the identical cages without tail suspension. After 28 days of tail suspension, the animals were anesthetized by intraperitoneal injection of 10% chloral hydrate (2.5 ml/kg) and sacrificed by cervical dislocation. All efforts were made to minimize the discomfort of the animals.

## Transmission electron microscopy

After 28 days of tail suspension, the rat was euthanized by cervical dislocation and the brain was removed. The hippocampus tissue was separated on ice and cut into approximately $1mm^3$ tissue pieces, which were quickly placed in 2.5% glutaraldehyde fixative solution and fixed at 4˚C overnight. Samples were rinsed three times with phosphate buffer (0.1M, pH 7.0), then fixed with 1% osmium acid solution for 2h, rinsed again with phosphate buffer for three times, and dehydrated with gradient ethanol, put in epoxy resin, treated at 80˚C for 24h to polymerize, then cut into 100nm ultra-thin sections and dyed with uranyl acetate and lead citrate. Finally it was observed under a transmission electron microscope (Hitachi, Japan). The number of mitochondria in each neuronal soma was counted from at least 10 cells in each group. The micrographs were tracked respectively, and the shape and size parameters were obtained by ImageJ software. The surface area representing the size of mitochondria was reported in squared micrometers ($\mu m^2$), and Ferret's diameter was measured by the longest distance between two points within a mitochondria.

## Isolation of mitochondria

Fifteen rats were randomly taken out from the tail suspension and control groups. After separating the hippocampus, the left and right hippocampus of five randomly selected rats were mixed together to form a sample. The mitochondria were extracted by Mitochondrial Isolation Kit (MP-007, Invent, USA) according to the manufacture's instructions. Protein concentrations were quantified by BCA assay (Thermo Fischer, USA).

## TMT-based proteomics analysis

**Protein processing.** Dissolved the mitochondria with Protein Solubilization Reagent for MS (WA-011, Invent, USA). Added the DTT to the sample to a final concentration of 10mM and reduced in the oven at 55˚C for 1h. After the samples returning to the room temperature, added the IAM at a final concentration of 40mM, and reacted for 30min in dark. Centrifuged with a 10kda ultrafiltration tube at 12000g, and added 200μl of 100mM TEAB to the samples, then added trypsin according to 1/50 of the protein mass, and kept in a water bath at 37˚C overnight. The next day, washed 3 times with ultrapure water and freeze-dried at the bottom of the enrichment tube.

**TMT labelling.** Re-dissolved the peptides with 200μl of 200mM TEAB, and then quantified them with NanoDrop (Thermo Fisher, USA). Taking 25μg of peptides from each sample and labelled them with TMT reagent at room temperature for 1 hour. The labeled samples were as follows: the Control group, labeled with 126, 127N and 128N; the SM group, labeled with 129N, 130N and 131. Then added 5μl of 5% ammonia water for quenching, mixed and reacted at room temperature for 15 minutes, and then mixed 5 labeled samples from same group together and evaporated to dryness at 60˚C by a concentrator.

**HPLC liquid phase separation.** Prepared mobile phase A liquid (2% acetonitrile, 98% water, ammonia water adjusted to pH = 10) and B liquid (98% acetonitrile, 2% water, ammonia water adjusted to pH = 10). Used 150μl of solution A to dissolve the lyophilized powder of the sample, centrifuged at 12000g for 10min at room temperature, and took the supernatant for injection. The TMT labelled peptide mixture was fractionated by a Durashell column from Agela (4.6mm×250mm i.d, C18, 5μm) by L-3000 HPLC system (Rigol, China). A total of 100 tubes were collected with the speed of 1 tube per minute, and finally combined into 10 fractions. All fractions were dried by a rotary vacuum concentrator.

**LC-MS/MS analysis.** The TMT-labeled sample of each fraction was resuspended in 2% acetonitrile, 98% water and 0.1% FA, after centrifuged at 12000g for 3min, 10μl of supernatant was loaded onto an Eksigent Nano LC 2D plus HPLC by the autosampler onto a 5μm C18 trap column (ID100μm, 20mm length). The peptides were then eluted onto a 3μm analytical C18 column (ID75μm, 120mm length) packed in-house. Separation was run at 330 nl/min starting from 5% B2 (98% ACN+1.9% H2O+0.1% FA), followed by stepwise gradient (8% B2 for 5min, 22% B2 for 34min, 32% B2 for 41min, 90% B2 for 42min), maintained at 90% B2 for another 46min, and finally returned to 5% B2 for 1min. LC-MS/MS analysis was performed on a Q Exactive HF mass spectrometer (Thermo Fisher, USA). The mass spectrometry parameters were set as follows: the first-level M/Z scan range was 300–1400, the resolution was 120,000, the AGC target was 3e6, the maximum ion injection time was 80ms, 15 precursor ions were selected for secondary fragmentation, and the secondary resolution was 60000. The AGC target was 5e4, the maximum ion injection time was 20ms, and the precursor ion window was set to 1.2M/Z. Proteome Discoverer (version 2.3.1.15, Thermo Fisher Scientific, USA) was used for data retrieval of raw data. The database used was downloaded from uniprotKB (including TrEMBL entries) on August 13, 2020, containing 20289 sequence information, and was also downloaded from Uniprot_Rat (version 2019.04.20). Andromeda search engine was used with the following settings: trypsin cleavage; fixed modification of carbamidomethylation of cysteine; variable modifications of oxidation of methionine; acetylation modification at the N-terminal of protein, the primary mass error was set at 20ppm and the secondary mass error was set at 20mmu; a maximum of two missed cleavage. The false discovery rate was calculated by decoy database searching. For protein identification, the peptides were of minimum 6 amino acids and had at least 1 unique peptide identified per protein. A false discovery rate (FDR) of 1% at both peptide and protein level was used. Normalization was performed against the total peptide amount.

## Analysis of mitochondrial proteome

In order to determine the possible biological functions of DEPs, DAVID Bioinformatics Resources were used to conduct gene ontology (GO) analysis in biological processes, cell components and molecular functions. The Kyoto Encyclopedia of Genes and Genomes (KEGG: http://www.genome.jp/kegg/) analysis was also conducted. The protein-protein interaction (PPI) network was constructed using the STRING database and Cytoscape.

## Western blotting

Mitochondria extraction and dissolution were conducted as described above, and the protein concentration was determined with BCA Protein Assay Kit (Thermo Fisher, USA). Western blot analyses were performed according to standard procedures. The primary antibodies used were aconitate hydratase (ACO2, 6571), dihydrolipoamide S-succinyltransferase (DLST, 11954), citrate synthase (CS, 14309), and VDAC (4661), and all of the antibodies were purchased from Cell Signaling Technology, Danvers, MA, USA. The immunoblot images were

analyzed with ImageJ to determine the relative integrated density, and the relative expressions of ACO2, DLST and CS were represented by the intensity ratio between the interested protein and the loading control (VDAC) in each loading lane.

## Statistical analysis

The data were expressed as mean±standard deviation (SD) or fold change (FC) relative to the corresponding control group. Statistical analysis was performed using two tailed Student's t-test. The Fisher's exact test was used to evaluate the significance of GO terms and Pathway enrichment, with correction for multiple comparisons based on the false discovery rate (FDR). A p-value or FDR less than 0.05 were considered statistically significant.

## Results

### Mitochondrial dynamics and morphology are changed by 28-day SM

We studied the influence of SM on the dynamics and morphology of mitochondria in hippocampal neuronal soma using transmission electron microscopy. It was found that the mitochondrial swelling was obvious, and the cristae were loose and dissolved in the SM group, while the mitochondrial structure in the Control group was basically normal, the membrane shape was complete, and the cristae were dense and regular (**Fig 1A–1D**). The number of mitochondria in the SM group was significantly increased compared to the Control group (p<0.001), which was reflected by the elevated surface area and the lengthened Feret's diameter (**Fig 1E–1G**). These data suggest that the function of mitochondria in the rat hippocampal neuronal soma is changed under the 28-day SM condition.

### The mitochondrial proteome of hippocampus is altered by SM

To evaluate the changes in hippocampus mitochondrial protein expression, we performed TMT-based proteomics analysis on the SM group and Control group samples. The heatmap plot disclosed that a total of 4,044 proteins were quantified across all samples, and the hippocampus mitochondrial protein expression was altered significantly by SM (**Fig 2A**). A fold change cutoff value of ≥1.5 or ≤0.67 was defined as up- or down-regulation, and only proteins that were identified by three or more peptides with >1.5-fold changes and statistically significant (p value≤0.05) were considered to be DEPs. Among the 163 DEPs, 128 proteins were upregulated in the SM group compared with the Control group, whereas 35 proteins were downregulated (**Fig 2B**). The gene and protein names, fold change and p-value of DEPs were listed in **Table 1** and **S1 Table**.

### Bioinformatics analysis of DEPs in mitochondrial proteome

In order to have a functional overview of the DEPs, we performed function annotation by GO and KEGG analyses. The most relevant and significant enriched terms and pathways (FDR<0.01) were illustrated by biological process (BP) (Table 2), cellular compartment (CC) (Table 3), molecular function (MF) (Table 4), and KEGG pathway (Table 5) separately. The GO and KEGG Term ID, Term, Rich factors (Ratio), Enrichment, FDR and Protein IDs included were all listed in these tables. The most significantly enriched GO terms in the biological process category were mainly annotated with the terms tricarboxylic acid (TCA) cycle (GO: 0006099, 13 proteins), fatty acid beta-oxidation using acyl-CoA dehydrogenase (GO: 0033539, 7 proteins), 2-oxoglutarate metabolic process (GO: 0006103, 5 proteins), etc. DEPs classified in the cellular component category were mainly annotated with the terms mitochondrion (GO: 0005739, 75 proteins) and mitochondrial matrix (GO: 0005759, 28 proteins). DEPs

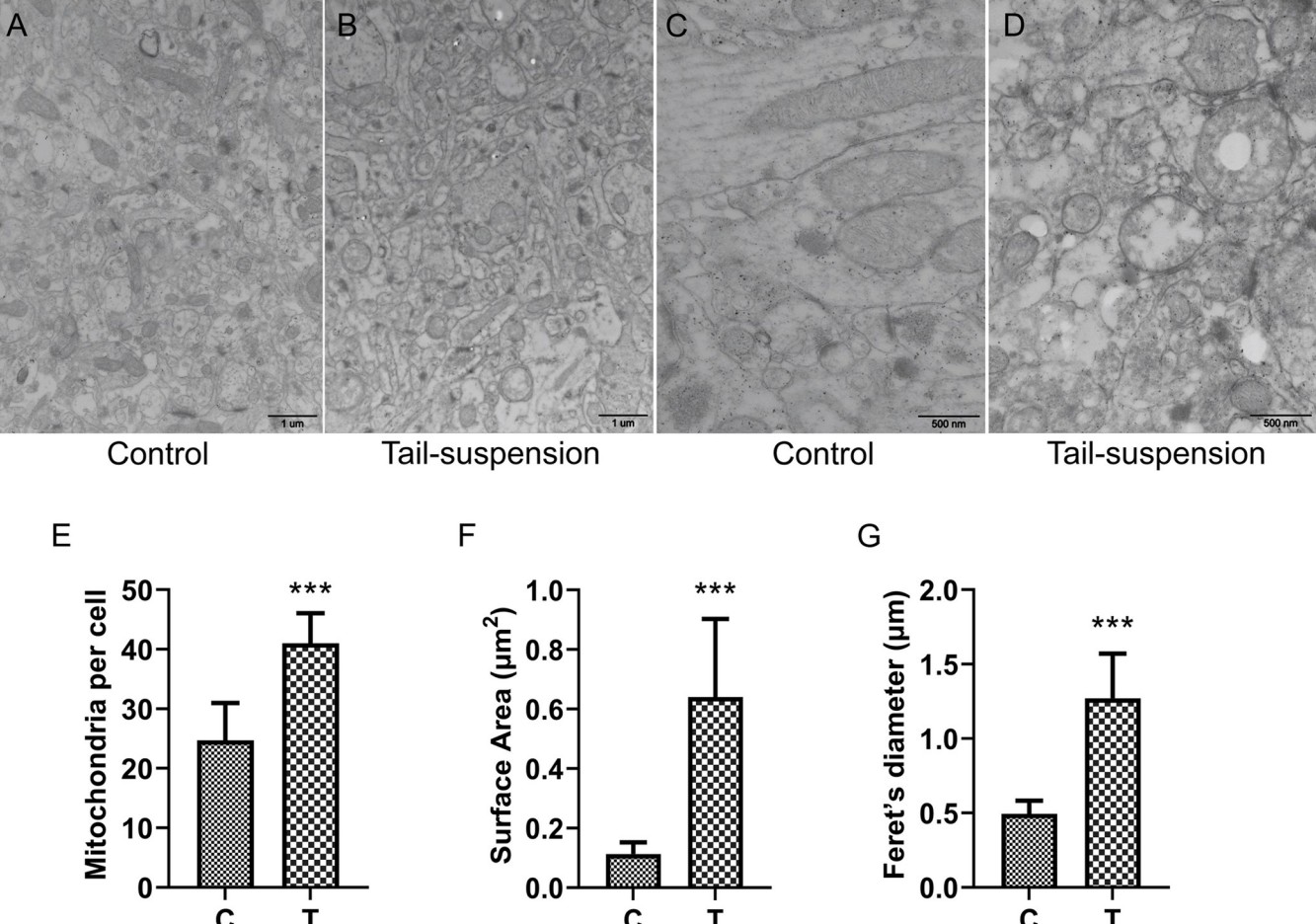

**Fig 1. Effects of SM on mitochondrial morphology and dynamics of hippocampal neuronal soma in rats.** Rats were tail suspended for 28 days, and the mitochondria in hippocampal neuronal soma were observed under transmission electron microscope. **(A, B)** Zoom 20,000 times. n = 6. **(C, D)** Zoom 50,000 times. **(E)** Mitochondria per cell were counted. n≥10 images per group. **(F, G)** mitochondrial mean surface area and Feret's diameter on transmission electron microscopy images. n = 25. C and T represented the Control group and SM group samples respectively. ***p<0.001.

classified in the molecular function category were mainly annotated with the terms flavin adenine dinucleotide binding (GO: 0050660, 8 proteins), pyridoxal phosphate binding (GO: 0030170, 8 proteins), NAD binding (GO: 0051287, 8 proteins) and Succinate-CoA ligase (ADP-forming) activity (GO: 0004775, 3 proteins). The 163 DEPs were annotated with KEGG pathways, and the top 3 most enriched pathways were carbon metabolism (path: rno01200, 19 proteins), valine, leucine and isoleucine degradation (path: rno00280, 12 proteins) and metabolic pathways (path: rno01100, 40 proteins), and the citrate cycle (TCA cycle) was one of the significantly enriched metabolic pathways.

Next, we constructed the protein-protein interaction (PPI) network to screen for hub proteins (**Fig 3**). The top 10 high-degree hub nodes included DLD (Dihydrolipoyl dehydrogenase), CS (Citrate synthase), ACO2 (Aconitate hydratase), MDH2 (Malate dehydrogenase), DLST (Dihydrolipoamide S-succinyltransferase), IDH3A (Isocitrate dehydrogenase [NAD] subunit), ALDH6A1 (Aldehyde dehydrogenase family 6, subfamily A1, isoform CRA_b), FH (Fumarate hydratase), GLUD1 (Glutamate dehydrogenase 1), and GOT2 (Aspartate aminotransferase), and they may play an important role in mediating the effects of SM on the

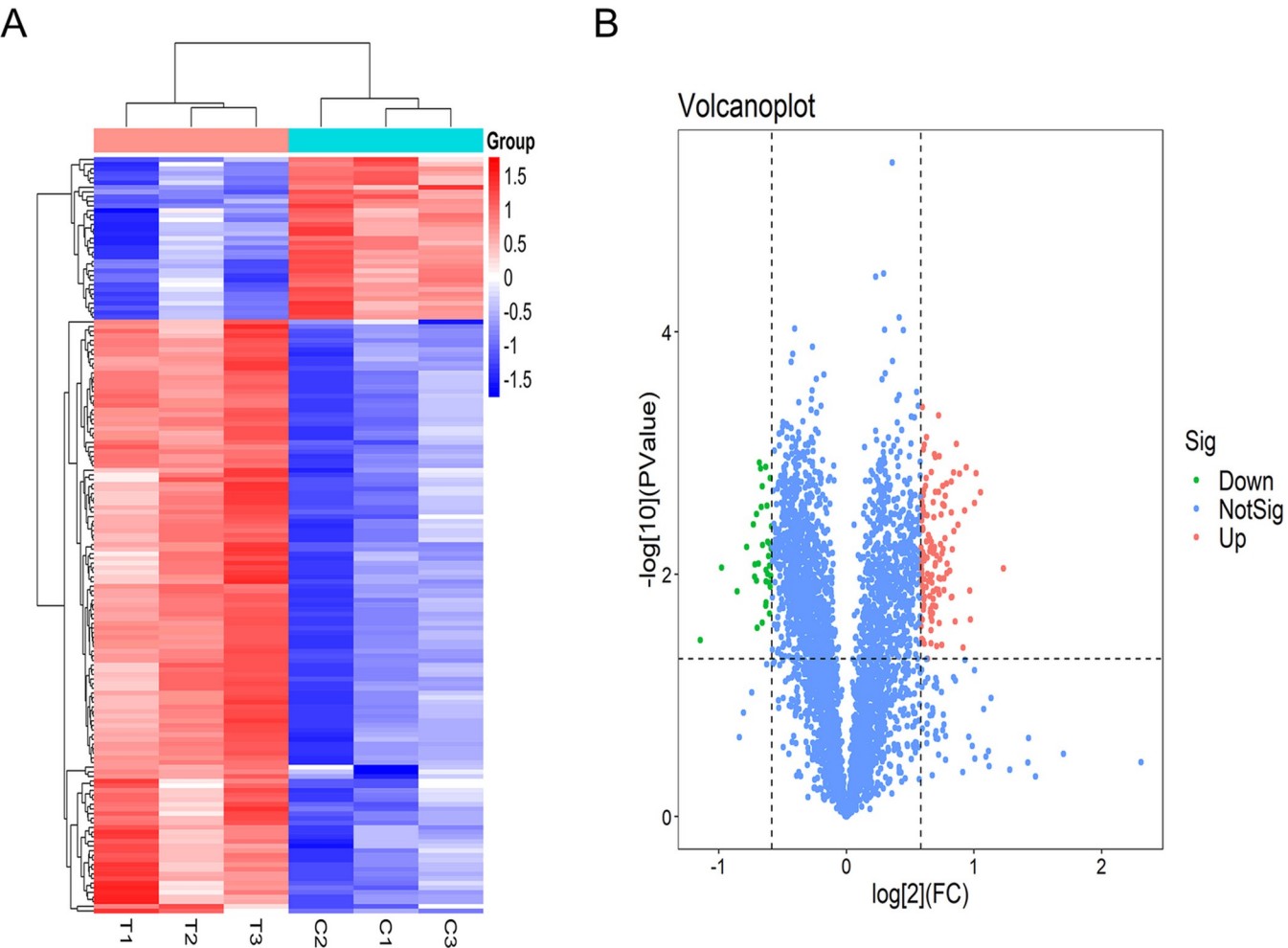

**Fig 2. The overview of global changes of identified proteins.** (A) The identified proteins expression profile by heatmap plot. T1, T2 and T3, and C1, C2 and C3 respectively represented 3 tail suspension samples and 3 control samples. (B) Volcano plots of fold-change vs. -log₁₀ p-value of identified proteins. The significance threshold was set at p-value≤0.05. The significant differentially expressed proteins were marked with different colors, the upregulated proteins were marked in pink, downregulated proteins were marked in green, and blue dots indicated no significant difference. FC, fold change.

mitochondrial metabolic function. These data indicate that the effects of SM on mitochondria are mainly in the pathway of material metabolism and energy metabolism.

## Validation of the selected proteins in TCA by Western blotting

As the tricarboxylic acid (TCA) cycle was the most enriched GO terms in the biological process category, we used Western blotting to verify the relative expressions of the 3 proteins, ACO2 DLST and CS. All of the 3 proteins were involved in TCA cycle and were the hub genes in PPI network we constructed. The mass spectrometry results showed that their mean expression levels reached 1.71 (ACO2), 1.53 (DLST), and 1.85 (CS) times that of the Control group (**Table 1**). The Western blotting results showed that after 28 days of tail suspension, all three proteins were significantly upregulated, and the relative expressions of ACO2, DLST and CS reached 4.31, 2.70 and 2.70 times that of the corresponding control group, respectively (**Fig 4**). This suggests that SM greatly promotes the function of TCA cycle.

**Table 1.  The protein changes of mitochondria of hippocampus after 28 days of tail suspension in rats.**

| ID | Accession | Gene name | Protein name | Fold change | p-value |
|---|---|---|---|---|---|
| A0A0G2JYW6_RAT | | | Uncharacterized protein | 2.35 | 0.008939 |
| HOT_RAT | Q4QQW3 | Adhfe1 | Hydroxyacid-oxoacid transhydrogenase, mitochondrial | 2.07 | 0.002111 |
| G3V8U8_RAT | G3V8U8 | Bcat2 | Branched-chain-amino-acid aminotransferase | 2.02 | 0.001467 |
| SODM_RAT | P07895 | Sod2 | Superoxide dismutase [Mn], mitochondrial | 2.01 | 0.00258 |
| ATP6_RAT | P05504 | Mt-atp6 | ATP synthase subunit a | 1.96 | 0.02367 |
| AATM_RAT | P00507 | Got2 | Aspartate aminotransferase, mitochondrial | 1.95 | 0.013689 |
| CH10_RAT | P26772 | Hspe1 | 10 kDa heat shock protein, mitochondrial | 1.92 | 0.001314 |
| GCSH_RAT | Q5I0P2 | Gcsh | Glycine cleavage system H protein, mitochondrial | 1.90 | 0.002974 |
| TRXR2_RAT | Q9Z0J5 | Txnrd2 | Thioredoxin reductase 2, mitochondrial | 1.88 | 0.040301 |
| G3V936_RAT | G3V936 | Cs | Citrate synthase | 1.85 | 0.00148 |
| D3ZT98_RAT | D3ZT98 | Bola3 | BolA family member 3 | 1.83 | 0.003921 |
| GABT_RAT | P50554 | Abat | 4-aminobutyrate aminotransferase, mitochondrial | 1.82 | 0.000839 |
| A0A0G2K2Q2_RAT | A0A0G2K2Q2 | Gcat | Glycine C-acetyltransferase | 1.81 | 0.024635 |
| AUHM_RAT | F1LU71 | Auh | Methylglutaconyl-CoA hydratase, mitochondrial | 1.79 | 0.004409 |
| ES1_RAT | P56571 | | ES1 protein homolog, mitochondrial | 1.79 | 0.001447 |
| FAHD1_RAT | Q6AYQ8 | Fahd1 | Acylpyruvase FAHD1, mitochondrial | 1.78 | 0.006231 |
| G3V945_RAT | G3V945 | Aldh5a1 | Succinate-semialdehyde dehydrogenase | 1.77 | 0.015407 |
| A0A0G2JVW3_RAT | A0A0G2JVW3 | Ankrd17 | Ankyrin repeat domain 17 | 1.77 | 0.015563 |
| IDHP_RAT | P56574 | Idh2 | Isocitrate dehydrogenase [NADP], mitochondrial | 1.77 | 0.003087 |
| DHE3_RAT | P10860 | Glud1 | Glutamate dehydrogenase 1, mitochondrial | 1.77 | 0.001715 |
| F1LN88_RAT | F1LN88 | Aldh2 | Aldehyde dehydrogenase, mitochondrial | 1.75 | 0.00937 |
| IVD_RAT | P12007 | Ivd | Isovaleryl-CoA dehydrogenase, mitochondrial | 1.73 | 0.008189 |
| THTR_RAT | P24329 | Tst | Thiosulfate sulfurtransferase | 1.73 | 0.009805 |
| ECHM_RAT | P14604 | Echs1 | Enoyl-CoA hydratase, mitochondrial | 1.72 | 0.001788 |
| ACON_RAT | Q9ER34 | Aco2 | Aconitate hydratase, mitochondrial | 1.71 | 0.002251 |
| CALR_RAT | P18418 | Calr | Calreticulin | 1.71 | 0.00505 |
| PPIF_RAT | P29117 | Ppif | Peptidyl-prolyl cis-trans isomerase F, mitochondrial | 1.70 | 0.002718 |
| B2RYT0_RAT | B2RYT0 | Mrps21 | Mitochondrial ribosomal protein S21 | 1.70 | 0.010777 |
| D4A5F4_RAT | D4A5F4 | RGD1311575 | Hypothetical LOC289568 | 1.70 | 0.013594 |
| ECH1_RAT | Q62651 | Ech1 | Delta(3,5)-Delta(2,4)-dienoyl-CoA isomerase, mitochondrial | 1.69 | 0.006191 |
| ETFA_RAT | P13803 | Etfa | Electron transfer flavoprotein subunit alpha, mitochondrial | 1.69 | 0.002644 |
| G3V9U2_RAT | G3V9U2 | Acaa2 | 3-ketoacyl-CoA thiolase, mitochondrial | 1.69 | 0.001865 |
| D4A7X5_RAT | D4A7X5 | Ppm1k | Protein phosphatase 1K (PP2C domain containing) (Predicted) | 1.68 | 0.006727 |
| KNG1_RAT | P08934 | Kng1 | Kininogen-1 | 1.68 | 0.038332 |
| G3V7I0_RAT | G3V7I0 | Prdx3 | Peroxiredoxin 3 | 1.67 | 0.003211 |
| F1LNF7_RAT | F1LNF7 | Idh3a | Isocitrate dehydrogenase [NAD] subunit, mitochondrial | 1.67 | 0.005636 |
| Q6IMX3_RAT | Q6IMX3 | Acads | Acetyl-Coenzyme A dehydrogenase, short chain, isoform CRA_a | 1.67 | 0.01089 |
| A0A0G2JVM0_RAT | A0A0G2JVM0 | Aldh4a1 | Delta-1-pyrroline-5-carboxylate dehydrogenase, mitochondrial | 1.66 | 0.007541 |
| D4A8N2_RAT | D4A8N2 | Fdx2 | Ferredoxin 2 | 1.66 | 0.0146 |
| A0A0G2JZA2_RAT | A0A0G2JZA2 | Grpel1 | GrpE protein homolog | 1.66 | 0.02531 |
| 3HIDH_RAT | P29266 | Hibadh | 3-hydroxyisobutyrate dehydrogenase, mitochondrial | 1.66 | 0.001062 |
| MESD_RAT | Q5U2R7 | Mesd | LRP chaperone MESD | 1.65 | 0.010713 |
| GATA_RAT | Q5FWT5 | Qrsl1 | Glutamyl-tRNA(Gln) amidotransferase subunit A, mitochondrial | 1.65 | 0.014454 |
| FUMH_RAT | P14408 | Fh | Fumarate hydratase, mitochondrial | 1.65 | 0.000491 |
| A0A0G2K9G3_RAT | A0A0G2K9G3 | Mrps24 | Mitochondrial ribosomal protein S24 | 1.65 | 0.00209 |
| D4ADD7_RAT | D4ADD7 | Glrx5 | Glutaredoxin 5 | 1.65 | 0.013459 |

*(Continued)*

**Table 1.** (Continued)

| ID | Accession | Gene name | Protein name | Fold change | p-value |
|---|---|---|---|---|---|
| Q6AXY8_RAT | Q6AXY8 | Dhrs1 | Dehydrogenase/reductase (SDR family) member 1 | 1.63 | 0.039263 |
| MDHM_RAT | P04636 | Mdh2 | Malate dehydrogenase, mitochondrial | 1.63 | 0.007061 |
| G3V7J0_RAT | G3V7J0 | Aldh6a1 | Aldehyde dehydrogenase family 6, subfamily A1, isoform CRA_b | 1.63 | 0.013342 |
| MAAI_RAT | P57113 | Gstz1 | Maleylacetoacetate isomerase | 1.63 | 0.018412 |
| ACADL_RAT | P15650 | Acadl | Long-chain specific acyl-CoA dehydrogenase, mitochondrial | 1.62 | 0.002101 |
| LYRM9_RAT | B2RZD7 | Lyrm9 | LYR motif-containing protein 9 | 1.62 | 0.001083 |
| THIL_RAT | P17764 | Acat1 | Acetyl-CoA acetyltransferase, mitochondrial | 1.62 | 0.008093 |
| MANF_RAT | P0C5H9 | Manf | Mesencephalic astrocyte-derived neurotrophic factor | 1.62 | 0.008521 |
| F1M5N4_RAT | F1M5N4 | Me3 | Malic enzyme | 1.62 | 0.005954 |
| D4A0Y4_RAT | D4A0Y4 | Oxnad1 | Oxidoreductase NAD-binding domain containing 1 (Predicted), isoform CRA_b | 1.61 | 0.001023 |
| A0A0A0MXW1_RAT | A0A0A0MXW1 | Bckdhb | 2-oxoisovalerate dehydrogenase subunit beta, mitochondrial | 1.61 | 0.006284 |
| G3V796_RAT | G3V796 | Acadm | Acetyl-Coenzyme A dehydrogenase, medium chain | 1.61 | 0.0034 |
| A0A0G2K5F1_RAT | A0A0G2K5F1 | Macrod1 | ADP-ribose glycohydrolase MACROD1 | 1.61 | 0.018189 |
| IDH3B_RAT | Q68FX0 | Idh3B | Isocitrate dehydrogenase [NAD] subunit beta, mitochondrial | 1.60 | 0.002549 |
| G3V7I5_RAT | G3V7I5 | Aldh1b1 | Aldehyde dehydrogenase X, mitochondrial | 1.60 | 0.006674 |
| D3ZKG1_RAT | D3ZKG1 | Mmut | Methylmalonyl CoA mutase | 1.60 | 0.015932 |
| F210A_RAT | Q5XIJ4 | Fam210a | Protein FAM210A | 1.60 | 0.013908 |
| FMT_RAT | Q5I0C5 | Mtfmt | Methionyl-tRNA formyltransferase, mitochondrial | 1.60 | 0.018123 |
| CEGT_RAT | Q9R0E0 | Ugcg | Ceramide glucosyltransferase | 1.60 | 0.020969 |
| A0A0H2UI21_RAT | A0A0H2UI21 | Crat | Carnitine O-acetyltransferase | 1.59 | 0.011738 |
| M0R3V4_RAT | M0R3V4 | Mydgf | Myeloid-derived growth factor | 1.59 | 0.010141 |
| OAT_RAT | P04182 | Oat | Ornithine aminotransferase, mitochondrial | 1.59 | 0.024142 |
| F1LPV8_RAT | F1LPV8 | Suclg2 | Succinate—CoA ligase [GDP-forming] subunit beta, mitochondrial | 1.59 | 0.005411 |
| A0A0G2JSS8_RAT | A0A0G2JSS8 | Prdx5 | Peroxiredoxin | 1.59 | 0.001438 |
| RCN2_RAT | Q62703 | Rcn2 | Reticulocalbin-2 | 1.59 | 0.028981 |
| ATIF1_RAT | Q03344 | Atp5if1 | ATPase inhibitor, mitochondrial | 1.58 | 0.037234 |
| Q5U3Z7_RAT | Q5U3Z7 | Shmt2 | Serine hydroxymethyltransferase | 1.58 | 0.022602 |
| SCOT1_RAT | B2GV06 | Oxct1 | Succinyl-CoA:3-ketoacid coenzyme A transferase 1, mitochondrial | 1.58 | 0.019463 |
| Q68FZ8_RAT | Q68FZ8 | Pccb | Propionyl coenzyme A carboxylase, beta polypeptide | 1.58 | 0.01069 |
| FAHD2_RAT | B2RYW9 | Fahd2 | Fumarylacetoacetate hydrolase domain-containing protein 2 | 1.58 | 0.006075 |
| F1M8H2_RAT | F1M8H2 | Wars2 | Tryptophanyl tRNA synthetase 2 (mitochondrial) | 1.58 | 0.020609 |
| Q6AY99_RAT | Q6AY99 | Akr1b10 | Aldo-keto reductase family 1 member B10 | 1.57 | 0.015156 |
| A0A0H2UHE1_RAT | A0A0H2UHE1 | Suclg1 | Succinate—CoA ligase [ADP/GDP-forming] subunit alpha, mitochondrial | 1.57 | 0.005037 |
| D4A830_RAT | D4A830 | Ppa2 | Pyrophosphatase (inorganic) 2 | 1.57 | 0.006482 |
| F1LP30_RAT | F1LP30 | Mccc1 | Methylcrotonoyl-CoA carboxylase subunit alpha, mitochondrial | 1.56 | 0.01137 |
| D3ZTR1_RAT | D3ZTR1 | Mrps17 | Mitochondrial ribosomal protein S17 | 1.56 | 0.004775 |
| Q5RJR9_RAT | Q5RJR9 | Serpinh1 | Serine (Or cysteine) proteinase inhibitor, clade H, member 1, isoform CRA_b | 1.55 | 0.049977 |
| G3V6T7_RAT | G3V6T7 | Pdia4 | Protein disulfide-isomerase A4 | 1.55 | 0.009949 |
| C1QBP_RAT | O35796 | C1qbp | Complement component 1 Q subcomponent-binding protein, mitochondrial | 1.55 | 0.00327 |
| D4A833_RAT | D4A833 | Mrps30 | Mitochondrial ribosomal protein S30 | 1.55 | 0.007282 |
| DLDH_RAT | Q6P6R2 | Dld | Dihydrolipoyl dehydrogenase, mitochondrial | 1.55 | 0.005384 |
| A0A0H2UI42_RAT | A0A0H2UI42 | Mrpl30 | 39S ribosomal protein L30, mitochondrial | 1.54 | 0.00074 |
| AL7A1_RAT | Q64057 | Aldh7a1 | Alpha-aminoadipic semialdehyde dehydrogenase | 1.54 | 0.012773 |
| HMCS2_RAT | P22791 | Hmgcs2 | Hydroxymethylglutaryl-CoA synthase, mitochondrial | 1.54 | 0.001604 |
| DHTK1_RAT | Q4KLP0 | Dhtkd1 | Probable 2-oxoglutarate dehydrogenase E1 component DHKTD1, mitochondrial | 1.54 | 0.001875 |
| F1LM47_RAT | F1LM47 | Sucla2 | Succinate—CoA ligase [ADP-forming] subunit beta, mitochondrial | 1.53 | 0.003233 |

(*Continued*)

**Table 1.** (Continued)

| ID | Accession | Gene name | Protein name | Fold change | p-value |
|---|---|---|---|---|---|
| M0R4L6_RAT | M0R4L6 | Gatb | Glutamyl-tRNA(Gln) amidotransferase subunit B, mitochondrial | 1.53 | 0.003219 |
| G3V6P2_RAT | G3V6P2 | Dlst | Dihydrolipoamide S-succinyltransferase (E2 component of 2-oxo-glutarate complex), isoform CRA_a | 1.53 | 0.015357 |
| A0A0G2JUZ5_RAT | A0A0G2JUZ5 | Gldc | Glycine cleavage system P protein | 1.53 | 0.037413 |
| CATB_RAT | P00787 | Ctsb | Cathepsin B | 1.53 | 0.000854 |
| G3V6F5_RAT | G3V6F5 | Elac2 | ElaC homolog 2 (E. coli) | 1.53 | 0.007722 |
| SDHF1_RAT | B0K036 | Sdhaf1 | Succinate dehydrogenase assembly factor 1, mitochondrial | 1.53 | 0.016151 |
| RM38_RAT | Q5PQN9 | Mrpl38 | 39S ribosomal protein L38, mitochondrial | 1.53 | 0.018071 |
| D3ZDX7_RAT | D3ZDX7 | Mrpl48 | Mitochondrial ribosomal protein L48 | 1.53 | 0.007002 |
| A0A0G2K7D7_RAT | A0A0G2K7D7 | Nars2 | Asparaginyl-tRNA synthetase 2, mitochondrial | 1.52 | 0.002014 |
| G3V879_RAT | G3V879 | Coq7 | 5-demethoxyubiquinone hydroxylase, mitochondrial | 1.52 | 0.010989 |
| ETFB_RAT | Q68FU3 | Etfb | Electron transfer flavoprotein subunit beta | 1.52 | 0.003036 |
| D3ZDP2_RAT | D3ZDP2 | Mrpl58 | Mitochondrial ribosomal protein L58 | 1.52 | 0.01954 |
| G3V8W9_RAT | G3V8W9 | Tstd3 | Similar to CG12279-PA | 1.52 | 0.000938 |
| D4AB01_RAT | D4AB01 | Hint2 | Histidine triad nucleotide binding protein 2 (Predicted), isoform CRA_a | 1.52 | 0.002216 |
| CH60_RAT | P63039 | Hspd1 | 60 kDa heat shock protein, mitochondrial | 1.52 | 0.004245 |
| D3ZUI9_RAT | D3ZUI9 | Ndufaf8 | NADH:ubiquinone oxidoreductase complex assembly factor 8 | 1.52 | 0.002759 |
| D3ZT90_RAT | D3ZT90 | Gcdh | Glutaryl-CoA dehydrogenase | 1.52 | 0.035278 |
| TM10C_RAT | Q5U2R4 | Trmt10c | tRNA methyltransferase 10 homolog C | 1.52 | 0.004136 |
| A0A0G2JW34_RAT | A0A0G2JW34 | Cisd3 | CDGSH iron sulfur domain 3 | 1.52 | 0.013893 |
| D3ZZR9_RAT | D3ZZR9 | Fkbp2 | Peptidylprolyl isomerase | 1.51 | 0.006559 |
| ACSF2_RAT | Q499N5 | Acsf2 | Acyl-CoA synthetase family member 2, mitochondrial | 1.51 | 0.004325 |
| D3ZJY1_RAT | D3ZJY1 | Mrpl28 | Mitochondrial ribosomal protein L28 | 1.51 | 0.012429 |
| PREY_RAT | Q5U1Z8 | Pyurf | Protein preY, mitochondrial | 1.51 | 0.02374 |
| M0RAK2_RAT | M0RAK2 | LOC684270 | RCG22622 | 1.51 | 0.000417 |
| RM10_RAT | P0C2C4 | Mrpl10 | 39S ribosomal protein L10, mitochondrial | 1.51 | 0.002504 |
| Q3MHT2_RAT | Q3MHT2 | Nfs1 | Cysteine desulfurase, mitochondrial | 1.51 | 0.0099 |
| SYDM_RAT | Q3KRD0 | Dars2 | Aspartate—tRNA ligase, mitochondrial | 1.51 | 0.009136 |
| COQ6_RAT | Q68FU7 | Coq6 | Ubiquinone biosynthesis monooxygenase COQ6, mitochondrial | 1.51 | 0.003601 |
| COX5B_RAT | P12075 | Cox5b | Cytochrome c oxidase subunit 5B, mitochondrial | 1.51 | 0.034268 |
| G3V828_RAT | G3V828 | Cnpy3 | Canopy FGF-signaling regulator 3 | 1.50 | 0.011124 |
| M0R7R2_RAT | M0R7R2 | LOC683897 | Similar to Protein C6orf203 | 1.50 | 0.008352 |
| FRDA_RAT | D3ZYW7 | Fxn | Frataxin, mitochondrial | 1.50 | 0.005993 |
| A0A0G2JTL5_RAT | A0A0G2JTL5 | Pc | Pyruvate carboxylase, mitochondrial | 1.50 | 0.007181 |
| F1LM33_RAT | F1LM33 | Lrpprc | Leucine-rich PPR motif-containing protein, mitochondrial | 1.50 | 0.002213 |
| A0A0A0MXZ0_RAT | A0A0A0MXZ0 | Isca1 | Iron-sulfur cluster assembly 1 homolog, mitochondrial | 1.50 | 0.026833 |
| IDHG1_RAT | P41565 | Idh3g | Isocitrate dehydrogenase [NAD] subunit gamma 1, mitochondrial | 1.50 | 0.002832 |
| SYGP1_RAT | Q9QUH6 | Syngap1 | Ras/Rap GTPase-activating protein SynGAP | 0.67 | 0.004016 |
| A0A0G2JZB8_RAT | A0A0G2JZB8 | Gpm6b | Neuronal membrane glycoprotein M6-b | 0.66 | 0.006135 |
| TBB2B_RAT | Q3KRE8 | Tubb2b | Tubulin beta-2B chain | 0.66 | 0.010335 |
| D4A1J3_RAT | D4A1J3 | Palm3 | Paralemmin 3 | 0.66 | 0.012666 |
| LDHA_RAT | P04642 | Ldha | L-lactate dehydrogenase A chain | 0.66 | 0.001603 |
| A0A096MJW6_RAT | A0A096MJW6 | Il1rapl1 | Interleukin-1 receptor accessory protein-like 1 | 0.66 | 0.021052 |
| A0A0G2K0M8_RAT | A0A0G2K0M8 | Ncam1 | Neural cell adhesion molecule 1 | 0.66 | 0.00849 |
| CLCB_RAT | P08082 | Cltb | Clathrin light chain B | 0.66 | 0.005589 |
| F1LR33_RAT | F1LR33 | Plppr2 | Phospholipid phosphatase-related protein type 2 | 0.66 | 0.011704 |

(*Continued*)

**Table 1.** (Continued)

| ID | Accession | Gene name | Protein name | Fold change | p-value |
|---|---|---|---|---|---|
| PALM_RAT | Q920Q0 | Palm | Paralemmin-1 | 0.66 | 0.007084 |
| F1MA89_RAT | F1MA89 | Ccny | Cyclin Y | 0.65 | 0.005391 |
| SV2B_RAT | Q63564 | Sv2b | Synaptic vesicle glycoprotein 2B | 0.65 | 0.009948 |
| SC6A1_RAT | P23978 | Slc6a1 | Sodium- and chloride-dependent GABA transporter 1 | 0.65 | 0.002724 |
| FGF14_RAT | Q8R5L7 | Fgf14 | Fibroblast growth factor 14 | 0.65 | 0.017128 |
| Q4V7D9_RAT | Q4V7D9 | Smpdl3b | Acid sphingomyelinase-like phosphodiesterase | 0.65 | 0.009225 |
| G3V7A9_RAT | G3V7A9 | Cldn10 | Claudin | 0.65 | 0.011512 |
| SCN2B_RAT | P54900 | Scn2b | Sodium channel subunit beta-2 | 0.65 | 0.001306 |
| D4AA77_RAT | D4AA77 | Plxnd1 | Plexin D1 | 0.64 | 0.018219 |
| A0A0G2JVB0_RAT | A0A0G2JVB0 | Slc2a3 | Solute carrier family 2, facilitated glucose transporter member 3-like | 0.63 | 0.005762 |
| HBA_RAT | P01946 | Hba1 | Hemoglobin subunit alpha-1/2 | 0.63 | 0.001886 |
| ENOB_RAT | P15429 | Eno3 | Beta-enolase | 0.63 | 0.025177 |
| GRID1_RAT | Q62640 | Grid1 | Glutamate receptor ionotropic, delta-1 | 0.63 | 0.002788 |
| F1M7N2_RAT | F1M7N2 | Entpd2 | Ectonucleoside triphosphate diphosphohydrolase 2 | 0.63 | 0.00134 |
| ALBU_RAT | P02770 | Alb | Serum albumin | 0.62 | 0.001197 |
| Q499T3_RAT | Q499T3 | Sirpa | Sirpa protein | 0.62 | 0.008158 |
| 2ABG_RAT | P97888 | Ppp2r2c | Serine/threonine-protein phosphatase 2A 55 kDa regulatory subunit B gamma isoform | 0.62 | 0.027676 |
| A0A0G2JTH4_RAT | A0A0G2JTH4 | Cd47 | Leukocyte surface antigen CD47 | 0.61 | 0.011316 |
| M0RBJ0_RAT | M0RBJ0 | Gng2 | Guanine nucleotide-binding protein subunit gamma | 0.61 | 0.003203 |
| G3V6R0_RAT | G3V6R0 | Slc1a2 | Amino acid transporter | 0.61 | 0.008327 |
| CRYM_RAT | Q9QYU4 | Crym | Ketimine reductase mu-crystallin | 0.61 | 0.010508 |
| NRN1_RAT | O08957 | Nrn1 | Neuritin | 0.60 | 0.003871 |
| GPM6A_RAT | Q812E9 | Gpm6a | Neuronal membrane glycoprotein M6-a | 0.58 | 0.005952 |
| KCIP2_RAT | Q9JM59 | Kcnip2 | Kv channel-interacting protein 2 | 0.55 | 0.013844 |
| F1M9G9_RAT | F1M9G9 | Scn2a | Sodium channel protein | 0.51 | 0.008815 |
| E9PSV8_RAT | E9PSV8 | Gpm6b | Neuronal membrane glycoprotein M6-b | 0.45 | 0.034939 |

## Discussion

In this study, we found that 28 days of tail suspension increased the number and size of mito-chondria in the hippocampus of rats and TMT-based proteomics analysis revealed 128 mito-chondrial proteins upregulation and 35 mitochondrial proteins downregulation. Bioinformatics analysis implied that mitochondrial metabolic pathways related TCA cycle and fatty acid oxidation were significantly changed. We verified the upregulated expressions of

**Table 2. The significantly enriched GO terms related to biological processes.**

| Term ID | Term | Ratio | Enrichment | FDR | Protein IDs |
|---|---|---|---|---|---|
| GO:0006099 | tricarboxylic acid cycle | 0.52 | 12.10 | 4.40E-10 | ACON_RAT,MDHM_RAT,F1LM47_RAT,F1LNF7_RAT,IDH3B_RAT, FUMH_RAT,IDHG1_RAT,G3V6P2_RAT,IDHP_RAT,G3V936_RAT, A0A0H2UHE1_RAT,F1LPV8_RAT,DHTK1_RAT |
| GO:0033539 | fatty acid beta-oxidation using acyl-CoA dehydrogenase | 0.78 | 8.44 | 1.01E-06 | ETFA_RAT,IVD_RAT,ACADL_RAT,ETFB_RAT,D3ZT90_RAT,G3V796_RAT, Q6IMX3_RAT |
| GO:0006103 | 2-oxoglutarate metabolic process | 0.63 | 5.40 | 0.000742 | AATM_RAT,DLDH_RAT,IDH3B_RAT,IDHG1_RAT,IDHP_RAT |
| GO:0006102 | isocitrate metabolic process | 0.80 | 4.99 | 0.001418 | ACON_RAT,IDH3B_RAT,IDHG1_RAT,IDHP_RAT |
| GO:0019254 | carnitine metabolic process, CoA-linked | 1.00 | 4.25 | 0.006217 | ACADL_RAT,A0A0H2UI21_RAT,G3V796_RAT |

**Table 3. The significantly enriched GO terms related to cellular component.**

| Term ID | Term | Ratio | Enrichment | FDR | Protein IDs | Accession |
|---|---|---|---|---|---|---|
| GO:0005739 | mitochondrion | 0.16 | 30.24 | 7.92E-29 | ACON_RAT,CH60_RAT,DHE3_RAT, A0A0G2JTL5_RAT,GABT_RAT,MDHM_RAT, AATM_RAT,THIL_RAT,DLDH_RAT, F1LM47_RAT,F1LNF7_RAT,G3V945_RAT, FUMH_RAT,IDHG1_RAT,SCOT1_RAT, G3V7J0_RAT,A0A0G2JSS8_RAT,G3V6P2_RAT, F1M5N4_RAT,IDHP_RAT,A0A0H2UHE1_RAT, G3V7I5_RAT,ETFA_RAT,AL7A1_RAT,IVD_RAT, ACSF2_RAT,ES1_RAT,3HIDH_RAT,AUHM_RAT, ACADL_RAT,G3V9U2_RAT,OAT_RAT, A0A0H2UI21_RAT,ECHM_RAT,F1LPV8_RAT, C1QBP_RAT,SODM_RAT,CH10_RAT, D4AB01_RAT,Q5U3Z7_RAT,ATIF1_RAT, MAAI_RAT,G3V7I0_RAT,ECH1_RAT, TRXR2_RAT,A0A0G2JUZ5_RAT,CATB_RAT, PPIF_RAT,DHTK1_RAT,D4ADD7_RAT, COQ6_RAT,FAHD1_RAT,GCSH_RAT, GATA_RAT,D3ZT90_RAT,G3V796_RAT, TM10C_RAT,HMCS2_RAT,D4A833_RAT, Q6IMX3_RAT,Q6AY99_RAT,D3ZUI9_RAT, F1M8H2_RAT,D3ZT98_RAT,PREY_RAT, FMT_RAT,FRDA_RAT,HOT_RAT, A0A0G2K2Q2_RAT,A0A0G2K7D7_RAT, M0R4L6_RAT,A0A0G2K9G3_RAT,F210A_RAT, G3V8U8_RAT,SDHF1_RAT | Q9ER34,P63039,P10860,A0A0G2JTL5, P50554,P04636,P00507,P17764,Q6P6R2, F1LM47,F1LNF7,G3V945,P14408,P41565, B2GV06,G3V7J0,A0A0G2JSS8,G3V6P2, F1M5N4,P56574,A0A0H2UHE1,G3V7I5, P13803,Q64057,P12007,Q499N5,P56571, P29266,F1LU71,P15650,G3V9U2,P04182, A0A0H2UI21,P14604,F1LPV8,O35796, P07895,P26772,D4AB01,Q5U3Z7,Q03344, P57113,G3V7I0,Q62651,Q9Z0J5, A0A0G2JUZ5,P00787,P29117,Q4KLP0, D4ADD7,Q68FU7,Q6AYQ8,Q5I0P2, Q5FWT5,D3ZT90,G3V796,Q5U2R4,P22791, D4A833,Q6IMX3,Q6AY99,D3ZUI9,F1M8H2, D3ZT98,Q5U1Z8,Q5I0C5,D3ZYW7, Q4QQW3,A0A0G2K2Q2,A0A0G2K7D7, M0R4L6,A0A0G2K9G3,Q5XIJ4,G3V8U8, B0K036 |
| GO:0005759 | mitochondrial matrix | 0.33 | 18.83 | 1.02E-17 | CH60_RAT,DHE3_RAT,GABT_RAT,MDHM_RAT, AATM_RAT,THIL_RAT,DLDH_RAT,SCOT1_RAT, G3V936_RAT,ETFA_RAT,IVD_RAT,ACADL_RAT, OAT_RAT,ETFB_RAT,ECHM_RAT,C1QBP_RAT, CH10_RAT,THTR_RAT,Q5U3Z7_RAT, F1LP30_RAT,A0A0G2JZA2_RAT,PPIF_RAT, D4ADD7_RAT,HMCS2_RAT,SYDM_RAT, F1M8H2_RAT,D4A7X5_RAT,SDHF1_RAT | P63039,P10860,P50554,P04636,P00507, P17764,Q6P6R2,B2GV06,G3V936,P13803, P12007,P15650,P04182,Q68FU3,P14604, O35796,P26772,P24329,Q5U3Z7,F1LP30, A0A0G2JZA2,P29117,D4ADD7,P22791, Q3KRD0,F1M8H2,D4A7X5,B0K036 |

three TCA cycle related proteins, ACO2, DLST and CS. Our study suggests that SM can cause mitochondrial dynamic and metabolic function related proteins changes, which may be one of the mechanisms of the effects of space microgravity on brain function.

Previously, Mikheeva et al. reported that after 30 days flight on the Bion-M1 biosatellite, the number and size of mitochondria in the soma of motoneurons and in axons coming from the vestibular structures increased in mouse [22]. Tan et al. showed that when compared with cells maintained under normal gravity, BL6-10 cells treated with simulating microgravity showed higher mitochondrial content and more abundant cytoplasmic mitochondria, and significantly reduced glycolytic metabolism [23]. In consistent with these findings, our study showed that

**Table 4. The significantly enriched GO terms related to molecular function.**

| Term ID | Term | Ratio | Enrichment | FDR | Protein IDs |
|---|---|---|---|---|---|
| GO:0050660 | flavin adenine dinucleotide binding | 0.35 | 5.70 | 0.000539659 | DLDH_RAT,ETFA_RAT,IVD_RAT,ACADL_RAT,TRXR2_RAT,D3ZT90_RAT, G3V796_RAT,Q6IMX3_RAT |
| GO:0030170 | pyridoxal phosphate binding | 0.30 | 5.11 | 0.00106165 | GABT_RAT,AATM_RAT,ALBU_RAT,OAT_RAT,Q5U3Z7_RAT, Q3MHT2_RAT,A0A0G2JUZ5_RAT,A0A0G2K2Q2_RAT |
| GO:0051287 | NAD binding | 0.26 | 4.62 | 0.002185681 | DLDH_RAT,F1LNF7_RAT,IDH3B_RAT,IDHG1_RAT,F1M5N4_RAT, IDHP_RAT,3HIDH_RAT,LDHA_RAT |
| GO:0004775 | succinate-CoA ligase (ADP-forming) activity | 1.00 | 4.16 | 0.004661018 | F1LM47_RAT,A0A0H2UHE1_RAT,F1LPV8_RAT |

**Table 5. The significantly enriched KEGG pathway from the DEPs.**

| Term ID | Term | Ratio | Enrichment | FDR | Protein IDs |
|---------|------|-------|-----------|-----|-------------|
| path: rno01200 | Carbon metabolism | 0.33 | 12.58 | 2.92E-11 | ACON_RAT,DHE3_RAT,MDHM_RAT,AATM_RAT,THIL_RAT,DLDH_RAT, F1LM47_RAT,IDH3B_RAT,G3V7J0_RAT,G3V6P2_RAT,F1M5N4_RAT,IDHP_RAT, Q68FZ8_RAT,ECHM_RAT,ENOB_RAT,Q5U3Z7_RAT,GCSH_RAT,G3V796_RAT, Q6IMX3_RAT |
| path: rno00280 | Valine, leucine and isoleucine degradation | 0.46 | 9.86 | 7.66E-09 | GABT_RAT,THIL_RAT,DLDH_RAT,SCOT1_RAT,G3V7J0_RAT,Q68FZ8_RAT, AL7A1_RAT,IVD_RAT,3HIDH_RAT,ECHM_RAT,G3V796_RAT,Q6IMX3_RAT |
| path: rno01100 | Metabolic pathways | 0.11 | 9.49 | 1.19E-08 | ACON_RAT,DHE3_RAT,GABT_RAT,MDHM_RAT,AATM_RAT,THIL_RAT, DLDH_RAT,F1LM47_RAT,G3V945_RAT,IDH3B_RAT,G3V7J0_RAT, G3V6P2_RAT,F1M5N4_RAT,IDHP_RAT,Q68FZ8_RAT,AL7A1_RAT, IVD_RAT,3HIDH_RAT,ACADL_RAT,OAT_RAT,LDHA_RAT,ECHM_RAT, COX5B_RAT,ENOB_RAT,THTR_RAT,Q5U3Z7_RAT,Q3MHT2_RAT,MAAI_RAT, DHTK1_RAT,COQ6_RAT,FAHD1_RAT,GCSH_RAT,GATA_RAT,D3ZT90_RAT, G3V796_RAT,Q6IMX3_RAT,Q6AY99_RAT,M0R4L6_RAT,CEGT_RAT,ATP6_RAT |
| path: rno00640 | Propanoate metabolism | 0.50 | 7.83 | 4.13E-07 | GABT_RAT,THIL_RAT,DLDH_RAT,F1LM47_RAT,G3V7J0_RAT,Q68FZ8_RAT, LDHA_RAT,ECHM_RAT,G3V796_RAT |
| path: rno00310 | Lysine degradation | 0.47 | 5.89 | 2.86E-05 | THIL_RAT,DLDH_RAT,G3V6P2_RAT,AL7A1_RAT,ECHM_RAT,DHTK1_RAT, D3ZT90_RAT |
| path: rno00380 | Tryptophan metabolism | 0.44 | 5.66 | 3.51E-05 | THIL_RAT,DLDH_RAT,G3V6P2_RAT,AL7A1_RAT,ECHM_RAT,DHTK1_RAT, D3ZT90_RAT |
| path: rno00630 | Glyoxylate and dicarboxylate metabolism | 0.44 | 5.66 | 3.51E-05 | ACON_RAT,MDHM_RAT,THIL_RAT,DLDH_RAT,Q68FZ8_RAT,Q5U3Z7_RAT, GCSH_RAT |
| path: rno00020 | Citrate cycle (TCA cycle) | 0.41 | 5.44 | 5.03E-05 | ACON_RAT,MDHM_RAT,DLDH_RAT,F1LM47_RAT,IDH3B_RAT,G3V6P2_RAT, IDHP_RAT |
| path: rno00650 | Butanoate metabolism | 0.46 | 5.06 | 0.000107806 | GABT_RAT,THIL_RAT,G3V945_RAT,SCOT1_RAT,ECHM_RAT,Q6IMX3_RAT |
| path: rno00071 | Fatty acid degradation | 0.32 | 4.57 | 0.000295395 | THIL_RAT,AL7A1_RAT,ACADL_RAT,ECHM_RAT,D3ZT90_RAT,G3V796_RAT, Q6IMX3_RAT |
| path: rno00620 | Pyruvate metabolism | 0.35 | 4.26 | 0.000551788 | MDHM_RAT,THIL_RAT,DLDH_RAT,F1M5N4_RAT,AL7A1_RAT,LDHA_RAT |
| path: rno00410 | beta-Alanine metabolism | 0.36 | 3.64 | 0.002124661 | GABT_RAT,G3V7J0_RAT,AL7A1_RAT,ECHM_RAT,G3V796_RAT |
| path: rno01210 | 2-Oxocarboxylic acid metabolism | 0.40 | 3.20 | 0.005443792 | ACON_RAT,AATM_RAT,IDH3B_RAT,IDHP_RAT |
| path: rno01200 | Carbon metabolism | 0.33 | 12.58 | 2.92E-11 | ACON_RAT,DHE3_RAT,MDHM_RAT,AATM_RAT,THIL_RAT,DLDH_RAT, F1LM47_RAT,IDH3B_RAT,G3V7J0_RAT,G3V6P2_RAT,F1M5N4_RAT,IDHP_RAT, Q68FZ8_RAT,ECHM_RAT,ENOB_RAT,Q5U3Z7_RAT,GCSH_RAT,G3V796_RAT, Q6IMX3_RAT |

the mitochondrial number and size of rat hippocampus were increased after 28 days of tail suspension. As a highly dynamic organelle, the function of mitochondria is dynamically regulated by the fission and fusion in various cell types, thus regulating the morphology, quantity, distribution, metabolism and biogenesis of mitochondria [17]. Some studies indicated that mitochondrial division can enhance its function. Fulghum and Hill found that catecholamines promotes mitochondrial fission and up-regulates PGC1α, thereby dramatically increasing mitochondrial function and long-term increase in mitochondrial abundance and fatty acid oxidation capacity [24]. Rana et al. found that promoting mitochondrial fission in midlife Drosophila can improve multiple markers of mitochondrial function and reduce mitochondrial ROS levels [25]. In current study, the analysis of DEPs revealed that most of the proteins (128/163) were upregulated after tail suspension, suggesting that SM may enhance the function of mitochondria in the hippocampus possibly through influencing the fission of mitochondria, and this mechanism requires further investigation.

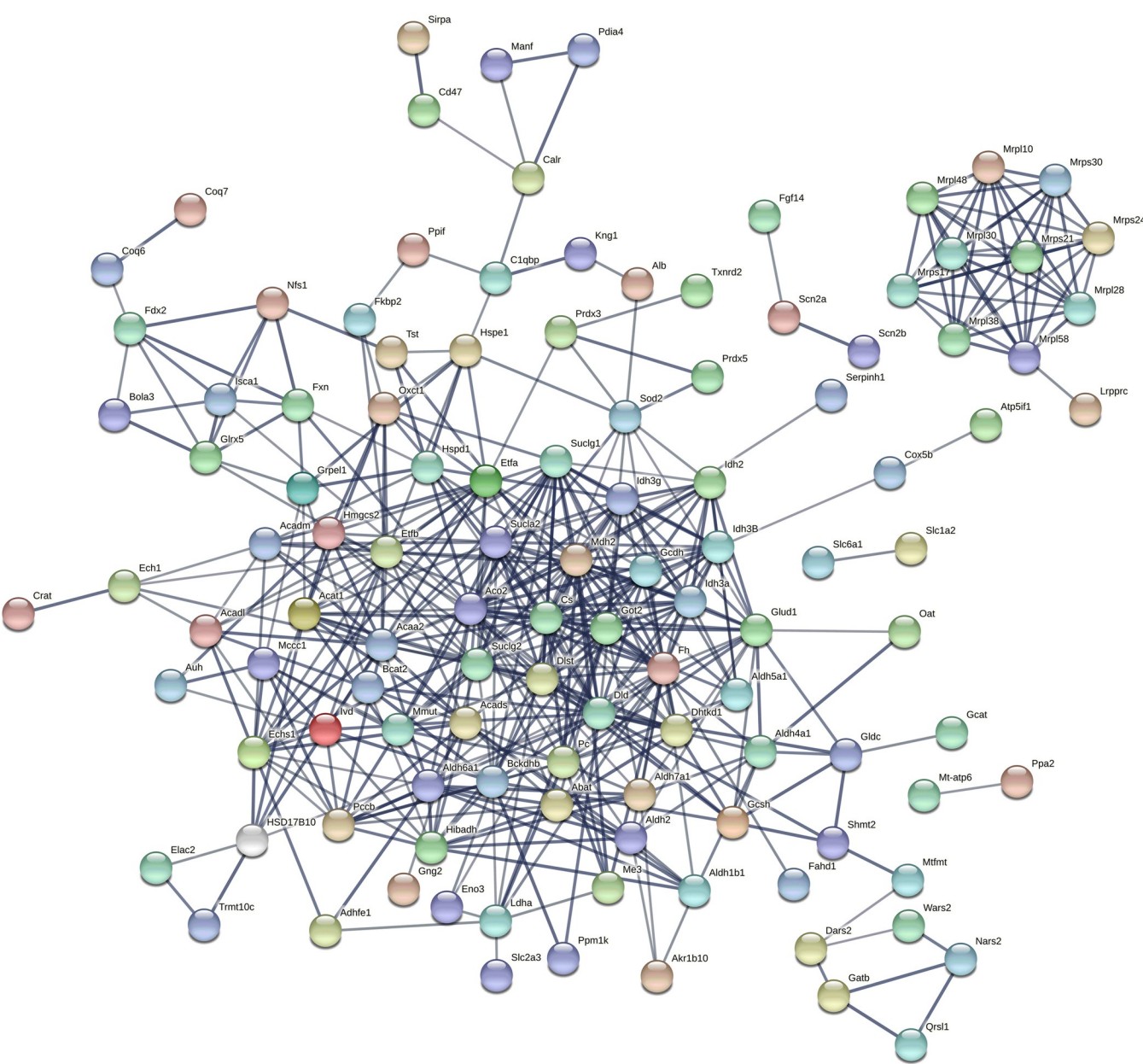

**Fig 3. The protein-protein interaction (PPI) network.** The colored nodes represented query proteins and first shell of interactors, whereas, white nodes represented second shell of interactors. Lines represented the interactions between two nodes.

The GO analysis of our current study showed that the most significantly enriched category in the cellular component were TCA cycle, fatty acid beta-oxidation using acyl-CoA dehydrogenase, 2-oxoglutarate metabolic process, isocitrate metabolic process and carnitine metabolic process, CoA-linked, and all of them are the important processes of mitochondria. These findings suggested that the DEPs are mainly involved in mitochondria and energy metabolism. In the TCA cycle pathway, all 13 identified proteins were upregulated. Similarly, the DEPs that involved in fatty acid beta-oxidation using acyl-CoA dehydrogenase process were also upregulated. Espinosa-Jeffrey et al. used 3D-clinostat robot to simulate the microgravity in oligodendrocytes, and found that the mitochondrial respiration and glycolysis are increased after 24

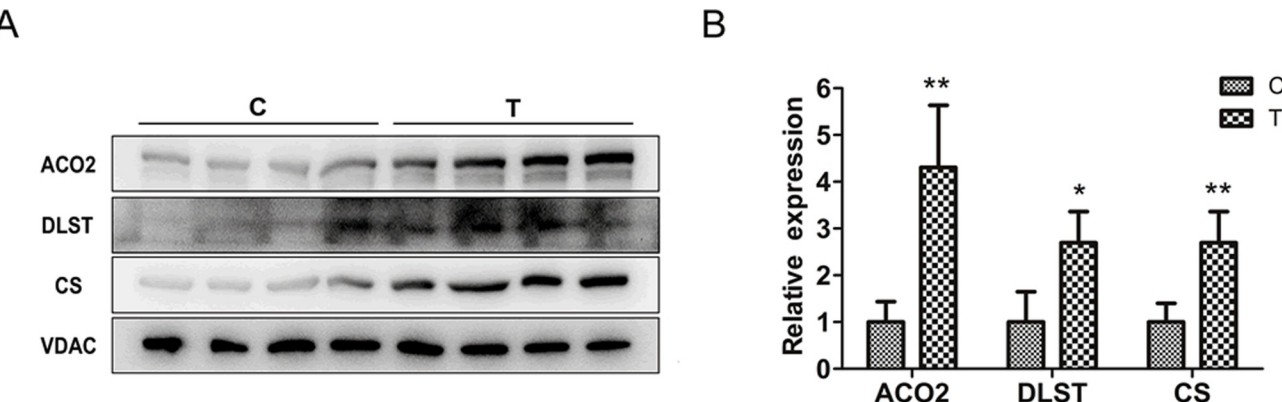

**Fig 4. 28-day tail suspension promotes the expression of ACO2, DLST and CS in hippocampus mitochondria of rats.** (A) Protein levels of ACO2, DLST and CS were determined by immunoblotting. (B) The relative expressions of ACO2, DLST and CS were represented by the intensity ratio between the protein and the loading control (VDAC) in each lane. n = 4, error bars indicated standard deviations. *p<0.05; **p<0.01 compared with each control group.

hours exposure to SM, indicating that SM enhances the mitochondrial function [26]. In another study, the primary osteoblasts were exposed to SM for 110 hours, and the metabonomics and proteomics results showed that TCA cycle is activated and acetyl coenzyme A is accumulated [27]. The real flight data also showed that microgravity increases the TCA activity. da Silveira et al. analyzed four human cell lines (fibroblasts, endothelial cells, primary T cells, and hair follicles) in vitro datasets available on GeneLab by gene set enrichment analysis (GSEA) for the overlapping pathways, and found one overlapping collection of gene sets across all four cell types, which contains four mitochondrial function gene ontology (GO) terms: mitochondrial ATP synthesis, mitochondrial electron transport, oxidative phosphorylation (OXPHOS), and hydrogen ion transmembrane transportation. They next analyzed the metabonomics data of gastrocnemius and quadriceps femoris muscles in mice after 35 days spaceflight, and the enrichment analysis showed that spaceflight increases the mitochondrial and energy metabolism related pathways, such as the b-oxidation of long-chain fatty acids and the TCA cycle [14]. The famous NASA twins study demonstrated that the levels of plasma TCA cycle intermediates (citric acid and malic acid) are raised during flight compared with pre-flight and post-flight levels [28], indicating that the TCA activity is elevated. Our results suggested that mitochondrial activity and energy metabolism are remarkably upregulated in rat hippocampus after 28-day tail suspension. Although the organizations studied are not the same, their results confirmed that our findings are similar to those in the real space microgravity environment. Material metabolism and energy metabolism are the main functions of mitochondria. The KEGG pathway analysis of our results indicated that 14 metabolic pathways showed significant differences between Control group and SM group. Almost all the material and energy metabolism, including carbon metabolism, amino acid metabolism (e.g., valine, leucine and isoleucine degradation, lysine degradation, tryptophan metabolism and beta-Alanine metabolism), TCA cycle and lipid metabolism (e.g., fatty acid degradation), have been changed significantly in our study, further proving that tail suspension can change the function of hippocampal mitochondria in rats.

Protein-protein interaction (PPI) research can reveal the protein function of DEPs at the molecular level, and explain the cellular mechanism by elucidating the interaction of whole genome proteins [29]. In this study, we constructed the PPI network of DEPs, and found that 6 (CS, ACO2, MDH2, DLST, IDH3A and FH) of top 10 high-degree hub nodes were involved in TCA cycle. In particular, we used Western blotting to detect the expressions of ACO2, CS

and DLST, and found that their expression trends were consistent with the results of proteomics. Considering that ACO2 belongs to aconitase family and plays an important role in maintaining oxidative phosphorylation and energy generation [30, 31], we believe that other nodal genes are also involved in the regulation of mitochondrial function or the downstream gene expression or metabolism by microgravity. Our proteomics analysis also showed that the expressions of antioxidant enzymes such as SOD2 and prdx3 were increased, suggesting that SM may induce the transfer of energy metabolism from glycolysis to oxidative phosphorylation in rat hippocampus. This may reflect the compensation of body to the harmful effects of SM.

The future success of long-term space exploration requires a comprehensive understanding of the impact of spaceflight on human biology. After analyzing the samples from 59 astronauts and hundreds of samples flown in space by transcriptomics, proteomics, metabolomics and epigenetics, da Silveira et al. concluded that mitochondrial disorders are the central hub of space biology [14]. In view of the fact that true microgravity cannot be simulated on the earth, the tail suspension model only simulate the fluid shift, muscle atrophy, bone loss effects in microgravity. In our study, the effects of SM on the dynamics and proteomics of mouse hippocampal mitochondria may be the result of long-term body fluid shift. In order to ensure the health of human spaceflight, more in-depth experimental research on mitochondrial functions and molecular mechanisms are needed in the aerospace environment.

## Supporting information

**S1 Fig. Uncropped images of Western blots in Fig 4.**
(TIF)

**S1 Table. Raw data of proteomics.**
(XLSX)

## Author Contributions

**Conceptualization:** Bo Song, Lina Qu.

**Data curation:** Guohua Ji, Hui Chang, Mingsi Yang, Hailong Chen, Lina Qu.

**Formal analysis:** Guohua Ji, Hui Chang, Hailong Chen, Bo Song, Lina Qu.

**Funding acquisition:** Guohua Ji, Yinghui Li, Bo Song, Lina Qu.

**Investigation:** Guohua Ji, Hui Chang, Mingsi Yang, Hailong Chen, Tingmei Wang, Xu Liu.

**Methodology:** Guohua Ji, Hui Chang, Ke Lv, Bo Song, Lina Qu.

**Project administration:** Yinghui Li, Lina Qu.

**Resources:** Yinghui Li, Lina Qu.

**Supervision:** Yinghui Li, Lina Qu.

**Validation:** Lina Qu.

**Visualization:** Guohua Ji, Lina Qu.

**Writing – original draft:** Guohua Ji, Hui Chang, Mingsi Yang, Bo Song, Lina Qu.

**Writing – review & editing:** Guohua Ji, Ke Lv, Yinghui Li, Bo Song, Lina Qu.

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
