## [Decision Letter · Decision Letter 0]

1 Dec 2021

PONE-D-21-31461The mitochondrial proteomic changes of rat hippocampus induced by 28-day simulated microgravityPLOS ONE

Dear Dr. Ji,

Thank you for submitting your manuscript to PLOS ONE. After careful consideration, we feel that it has merit but does not fully meet PLOS ONE’s publication criteria as it currently stands. Therefore, we invite you to submit a revised version of the manuscript that addresses the points raised during the review process.

Both reviewers appreciate rigor and results of your paper. However, some weaknesses and concerns were expressed by both referees mainly regarding the focus of the study and discussion of results. Since no functional analyses neither data on the morphology of mitochondria  are provided in support of proteomic data, authors are encouraged to correct the text or adequate the experimental design to support conclusions.Some experimental details should be better described in the Methods section. 

We look forward to receiving your revised manuscript.

Kind regards,

Angela Chambery, PhD

Academic Editor

PLOS ONE

Journal Requirements:

2. In your Methods section, please provide additional information on the animal research and ensure you have included details on : (1) methods of sacrifice (2) methods of anesthesia and/or analgesia, and (2) efforts to alleviate suffering.

[This work was supported by The Foundation of State Key Laboratory of Space Medicine Fundamentals and Application, China Astronaut Research and Training Center (SMFA17A03, SMFA19B02, SMFA19K08, SMFA17B09), The National Natural Science Foundation of China (81773930, 21635001, 31800707, 31800998). The funders had no role in the study design, data collection and analysis, decision to publish, and preparation of the manuscript.]

 [This work was supported by The Foundation of State Key Laboratory of Space Medicine Fundamentals and Application, China Astronaut Research and Training Center (SMFA17A03, SMFA19B02, SMFA19K08, SMFA17B09), The National Natural Science Foundation of China (81773930, 21635001, 31800707, 31800998). The funders had no role in the study design, data collection and analysis, decision to publish, and preparation of the manuscript. The funders had no role in study design, data collection and analysis, decision to publish, or preparation of the manuscript.]

Additional Editor Comments:

Two expert reviewers have assessed your submission and feel that it has potential for publication, and so I would like to invite you to revise the paper.

Both reviewers appreciate rigor and results of your paper. However, some weaknesses and concerns were expressed by both referees mainly regarding the focus of the study and discussion of results. Since no functional analyses neither data on morphology of mitochondria are provided in support of proteomic data, authors are encouraged to correct the text or adequate the experimental design to support conclusions. Some experimental details should be better described in the Methods section.

Reviewers' comments:

Reviewer's Responses to Questions

**Comments to the Author**

1. Is the manuscript technically sound, and do the data support the conclusions?

Reviewer #1: Partly

Reviewer #2: Partly

2. Has the statistical analysis been performed appropriately and rigorously? 

Reviewer #1: Yes

Reviewer #2: Yes

3. Have the authors made all data underlying the findings in their manuscript fully available?

Reviewer #1: Yes

Reviewer #2: Yes

4. Is the manuscript presented in an intelligible fashion and written in standard English?

Reviewer #1: Yes

Reviewer #2: Yes

5. Review Comments to the Author

Reviewer #1: Guohua Ji and coworkers present an original study focusing on the effects of simulated microgravity (by tail suspension for 28 days) on mitochondrial proteome in rat hippocampus.

The search for mitochondrial alterations in cognitive function decline caused by the aerospace microgravity environment represents the originality of the study.

The study was conducted with rigor and the results were well presented, however some weaknesses must be considered:

1) The authors claim to have obtained information on mitochondrial morphology and dynamics. However no morphometric analysis were done. Morphology should be “quantitatively” described by mitochondrial size measurements, reporting data on the mitochondrial mean surface area and Feret's diameter, for example.

2) The authors claim to have clearly observed in the hippocampus of T animals mitochondrial swelling, and cristae loos. This, together with an increase in mitochondrial number, lead the authors to state in favor of mitochondrial fission and altered mitochondrial dynamics. However, no data have been furnished on expression levels or activation of key markers of mitochondrial biogenesis and dynamics, such as PGC1α, mitofusins, OPA1 and DRP1.

So, conclusion on mitochondrial morphology and dynamics are not at all supported by the reported data and remain speculative.

3) Data obtained from the proteomic approach appear under-discussed. In particular, the observed up-regulation of key proteins of the tricarboxylic acid cycle seems to suggest a SM-induced energy shift from glycolysis to oxidative phosphorylation in the hippocampus of T rats. This, together with the observed mitochondrial ultrastructural alterations, could suggest a SM-induced mitochondrial damage. From this point of view, under-discussed is the reported up-regulation of antioxidant enzymes such as SOD2 and Peroxiredoxin 3.

4) The described structural alterations which population of mitochondria concern? The synaptic mitochondria? This is not clear.

5) In the manuscript test, the authors often talk about “mitochondrial function”, but no data have been furnished to conclude on respiratory properties, oxidative capacity, energy efficiency, ATP levels. They are encouraged to correct the text or adequate the experimental design to support conclusions.

6) Overall, the discussion is poor and not focalized.

7) As the authors themselves claim, the tail suspension model simulate fluid shift, muscle atrophy, bone loss, likely producing systemic metabolic adaptations. However, in the discussion, this aspect has been completely neglected even only as a framework for the possible effects of the model on the brain and in particular on the hippocampus.

Minor:

- in the text, some grammar correction is needed (Introduction: line 76, "are" is missing)

Reviewer #2: In this study, the authors describe changes occurring within the mitochondrial proteome of rat hippocampus by using the tail suspension microgravity simulation model. The work is well organized, however, some concerns should be addressed by the authors.

Minor revisions

Materials and methods section

- Please, detail the “Processing” and “Consensus” workflow nodes used for the raw data processing via Proteome Discoverer.

- Information on the chromatographic gradient used for LC-MS/MS analyses together with tags used for labelling are missing. Please, specify.

- Does the Uniprot_Rat database include TrEMBL entries? Please, specify by including the version of the UniProt release.

Results section

- In Table 1 and Table 3, please include the Accession Code of the identified proteins.

- In the PPI network, mark with a different colour additional interactors, besides DE identified proteins, if included.

Major revisions

Materials and methods section

The accurate sample quantification is a prerequisite for TMT labelling. Sample up- or under-estimation could tremendously affect the result of the analyses in terms of protein modulation. BCA assay is strongly recommended by ThermoFisher as detailed in the TMT datasheet. Since that it is not specified in the manuscript, did you perform it for samples quantification?

Discussion

The discussion of proteomic data is very poor. Moreover, besides indicating the pathways affected by changes in protein expression, the authors don’t provide data concerning neither the function nor the morphology of mitochondria.

Please, better focus the proteomic data discussion. In addition, attenuate the sentences concerning functional analyses if any additional experimental data are provided.

6. PLOS authors have the option to publish the peer review history of their article (what does this mean?). If published, this will include your full peer review and any attached files.

Reviewer #1: **Yes: **Elena Silvestri

Reviewer #2: No

---

## [Author Response · Author response to Decision Letter 0]

14 Jan 2022

Journal Requirements:

Reply: We confirmed that our manuscript meets PLOS ONE's style requirements.

2. In your Methods section, please provide additional information on the animal research and ensure you have included details on : (1) methods of sacrifice (2) methods of anesthesia and/or analgesia, and (2) efforts to alleviate suffering.

Reply: Information on (1) methods of sacrifice, (2) methods of anesthesia and/or analgesia, (3) efforts to alleviate suffering were added to the “Animals and SM model construction” and “Transmission electron microscopy” sections in the Material and Methods section.

[This work was supported by The Foundation of State Key Laboratory of Space Medicine Fundamentals and Application, China Astronaut Research and Training Center (SMFA17A03, SMFA19B02, SMFA19K08, SMFA17B09), The National Natural Science Foundation of China (81773930, 21635001, 31800707, 31800998). The funders had no role in the study design, data collection and analysis, decision to publish, and preparation of the manuscript.]

 [This work was supported by The Foundation of State Key Laboratory of Space Medicine Fundamentals and Application, China Astronaut Research and Training Center (SMFA17A03, SMFA19B02, SMFA19K08, SMFA17B09), The National Natural Science Foundation of China (81773930, 21635001, 31800707, 31800998). The funders had no role in the study design, data collection and analysis, decision to publish, and preparation of the manuscript. The funders had no role in study design, data collection and analysis, decision to publish, or preparation of the manuscript.]

Reply: Thanks for reminding. We have deleted the Acknowledgments section and the funding information in the manuscript and include the amended statements in the cover letter.

Reply: We have uploaded the original proteomic data as an attachment named S1_Raw_data. So the Data Availability statement can be changed to “All relevant data are within the manuscript and its Supporting Information files”.

Response: We have provided the original underlying images for Figure 4 in our manuscript with the name of S2_raw_images.

Additional Editor Comments:

Two expert reviewers have assessed your submission and feel that it has potential for publication, and so I would like to invite you to revise the paper.

Both reviewers appreciate rigor and results of your paper. However, some weaknesses and concerns were expressed by both referees mainly regarding the focus of the study and discussion of results. Since no functional analyses neither data on morphology of mitochondria are provided in support of proteomic data, authors are encouraged to correct the text or adequate the experimental design to support conclusions. Some experimental details should be better described in the Methods section.

Reply: Thank you. In order to make the article more rigorous, we have revised the discussion section to make it more focused; mitochondrial area and diameter were analyzed to support proteomic data; Part of the text has been corrected to support the conclusion; some experimental details are better described. Please refer to the revised manuscript for details.

Reviewers' comments:

Reviewer's Responses to Questions

Comments to the Author

1. Is the manuscript technically sound, and do the data support the conclusions?

Reviewer #1: Partly

Reviewer #2: Partly

2. Has the statistical analysis been performed appropriately and rigorously?

Reviewer #1: Yes

Reviewer #2: Yes

3. Have the authors made all data underlying the findings in their manuscript fully available?

Reviewer #1: Yes

Reviewer #2: Yes

4. Is the manuscript presented in an intelligible fashion and written in standard English?

Reviewer #1: Yes

Reviewer #2: Yes

5. Review Comments to the Author

Reviewer #1: Guohua Ji and coworkers present an original study focusing on the effects of simulated microgravity (by tail suspension for 28 days) on mitochondrial proteome in rat hippocampus.

The search for mitochondrial alterations in cognitive function decline caused by the aerospace microgravity environment represents the originality of the study.

The study was conducted with rigor and the results were well presented, however some weaknesses must be considered:

1) The authors claim to have obtained information on mitochondrial morphology and dynamics. However no morphometric analysis were done. Morphology should be “quantitatively” described by mitochondrial size measurements, reporting data on the mitochondrial mean surface area and Feret's diameter, for example.

Reply: Thank you for your suggestion. We analyzed the original picture of transmission electron microscope, and attached the results of the average surface area and Feret's diameter of mitochondria in the manuscript.

2) The authors claim to have clearly observed in the hippocampus of T animals mitochondrial swelling, and cristae loos. This, together with an increase in mitochondrial number, lead the authors to state in favor of mitochondrial fission and altered mitochondrial dynamics. However, no data have been furnished on expression levels or activation of key markers of mitochondrial biogenesis and dynamics, such as PGC1α, mitofusins, OPA1 and DRP1.

So, conclusion on mitochondrial morphology and dynamics are not at all supported by the reported data and remain speculative.

Reply: Thank you for your comments. We changed the word "fission" in the abstract and discussion section to "number" to make our conclusion more rigorous.

3) Data obtained from the proteomic approach appear under-discussed. In particular, the observed up-regulation of key proteins of the tricarboxylic acid cycle seems to suggest a SM-induced energy shift from glycolysis to oxidative phosphorylation in the hippocampus of T rats. This, together with the observed mitochondrial ultrastructural alterations, could suggest a SM-induced mitochondrial damage. From this point of view, under-discussed is the reported up-regulation of antioxidant enzymes such as SOD2 and Peroxiredoxin 3.

Reply: Thank you for your wonderful comments. We agree with you on energy shift and mitochondrial damage induced by SM, and modified the discussion part of the manuscript. Please refer to it. Although the ultrastructural changes of mitochondria and the up regulation of antioxidant enzymes such as SOD2 and peroxidase 3 were observed, there was no evidence that mitochondria were damaged. On the contrary, our results showed that the expression of metabolism related proteins were up-regulated, such as TCA cycle related enzymes, so we believe that tail suspension caused the up regulation of mitochondrial function.

4) The described structural alterations which population of mitochondria concern? The synaptic mitochondria? This is not clear.

Reply: The morphological changes of mitochondria come from the neuronal soma. We have changed the relevant expression in the manuscript to make it clearer. 

5) In the manuscript test, the authors often talk about “mitochondrial function”, but no data have been furnished to conclude on respiratory properties, oxidative capacity, energy efficiency, ATP levels. They are encouraged to correct the text or adequate the experimental design to support conclusions.

Reply: In order to make our conclusion more rigorous, we have changed the word “mitochondrial function” in line 305 to “mitochondrial metabolic function”, the “mitochondrial function” in line 344 to “function related proteins”, and the “function of mitochondria” in line 367 to “morphology of mitochondria” in the revised manuscript.

6) Overall, the discussion is poor and not focalized.

Reply: We have revised the discussion part of the manuscript according to the comments of the reviewers, and hope to meet your requirements.

7) As the authors themselves claim, the tail suspension model simulate fluid shift, muscle atrophy, bone loss, likely producing systemic metabolic adaptations. However, in the discussion, this aspect has been completely neglected even only as a framework for the possible effects of the model on the brain and in particular on the hippocampus.

Reply: In the discussion part, we explain the possible causes of the phenomena observed in this study.

Minor:

- in the text, some grammar correction is needed (Introduction: line 76, "are" is missing)

Reply: Thank you. We have corrected this mistake.

Reviewer #2: In this study, the authors describe changes occurring within the mitochondrial proteome of rat hippocampus by using the tail suspension microgravity simulation model. The work is well organized, however, some concerns should be addressed by the authors.

Minor revisions

Materials and methods section

- Please, detail the “Processing” and “Consensus” workflow nodes used for the raw data processing via Proteome Discoverer.

- Information on the chromatographic gradient used for LC-MS/MS analyses together with tags used for labelling are missing. Please, specify.

- Does the Uniprot_Rat database include TrEMBL entries? Please, specify by including the version of the UniProt release.

Reply: Thank you. We have described these methods in detail. Please refer to the TMT- based proteomics analysis section of the revised materials and methods.

Results section

- In Table 1 and Table 3, please include the Accession Code of the identified proteins.

- In the PPI network, mark with a different colour additional interactors, besides DE identified proteins, if included.

Reply: We have added Accession Code of the identified proteins to Table1 and Table 3. In addition to the DE identified protein, there is another protein involved in the second shell of interactors, which we have shown in the PPI network diagram with white colour.

Major revisions

Materials and methods section

The accurate sample quantification is a prerequisite for TMT labelling. Sample up- or under-estimation could tremendously affect the result of the analyses in terms of protein modulation. BCA assay is strongly recommended by ThermoFisher as detailed in the TMT datasheet. Since that it is not specified in the manuscript, did you perform it for samples quantification?

Reply: Thank you. The protein concentration was quantified by BCA assay and described in the method section of the revised manuscript.

Discussion

The discussion of proteomic data is very poor. Moreover, besides indicating the pathways affected by changes in protein expression, the authors don’t provide data concerning neither the function nor the morphology of mitochondria.

Please, better focus the proteomic data discussion. In addition, attenuate the sentences concerning functional analyses if any additional experimental data are provided.

Reply: Thank you for your comments. We supplemented the data of mitochondrial morphological analysis in the revised manuscript, and changed the word “mitochondrial function” to “mitochondrial number” in appropriate places; The discussion part has been improved to focus more on proteomic data.

6. PLOS authors have the option to publish the peer review history of their article (what does this mean?). If published, this will include your full peer review and any attached files.

Do you want your identity to be public for this peer review? For information about this choice, including consent withdrawal, please see our Privacy Policy.

Reviewer #1: Yes: Elena Silvestri

Reviewer #2: No

---

## [Decision Letter · Decision Letter 1]

27 Jan 2022

PONE-D-21-31461R1The mitochondrial proteomic changes of rat hippocampus induced by 28-day simulated microgravityPLOS ONE

Dear Dr. Ji,

Thank you for submitting your manuscript to PLOS ONE. After careful consideration, we feel that it has merit but does not fully meet PLOS ONE’s publication criteria as it currently stands. Therefore, we invite you to submit a revised version of the manuscript that addresses the points raised during the review process.

Expert reviewers have assessed your revised submission and feel that the paper has been greatly improved in terms of clarity of exposure, organization and responses to previous concerns.

Nevertheless, before acceptance, there are still few minor points that needs to be addressed regarding some typos or English grammar errors. We would like to invite you to carefully reread the text and correct any errors.

We look forward to receiving your revised manuscript.

Kind regards,

Angela Chambery, PhD

Academic Editor

PLOS ONE

Journal Requirements:

Additional Editor Comments (if provided):

Expert reviewers have assessed your revised submission and feel that the paper has been greatly improved in terms of clarity of exposure, organization and responses to previous concerns.

Nevertheless, before acceptance, there are still few minor points that needs to be addressed regarding some typos or English grammar errors. We would like to invite you to carefully reread the text and correct any errors.

Reviewers' comments:

Reviewer's Responses to Questions

**Comments to the Author**

1. If the authors have adequately addressed your comments raised in a previous round of review and you feel that this manuscript is now acceptable for publication, you may indicate that here to bypass the “Comments to the Author” section, enter your conflict of interest statement in the “Confidential to Editor” section, and submit your "Accept" recommendation.

Reviewer #1: All comments have been addressed

Reviewer #2: All comments have been addressed

2. Is the manuscript technically sound, and do the data support the conclusions?

Reviewer #1: Yes

Reviewer #2: Yes

3. Has the statistical analysis been performed appropriately and rigorously? 

Reviewer #1: Yes

Reviewer #2: N/A

4. Have the authors made all data underlying the findings in their manuscript fully available?

Reviewer #1: Yes

Reviewer #2: Yes

5. Is the manuscript presented in an intelligible fashion and written in standard English?

Reviewer #1: Yes

Reviewer #2: Yes

6. Review Comments to the Author

Reviewer #1: The authors answered the asked questions and accordingly modified the manuscript.

However, the text still contains some typos or English grammar errors.

Authors should carefully reread the text and correct any errors.

Reviewer #2: (No Response)

7. PLOS authors have the option to publish the peer review history of their article (what does this mean?). If published, this will include your full peer review and any attached files.

Reviewer #1: No

Reviewer #2: No

---

## [Author Response · Author response to Decision Letter 1]

10 Feb 2022

Journal Requirements:

Reply: We carefully checked the literatures cited in the revised manuscript, and found no retracted paper in the cited literatures.

Additional Editor Comments (if provided):

Expert reviewers have assessed your revised submission and feel that the paper has been greatly improved in terms of clarity of exposure, organization and responses to previous concerns.

Nevertheless, before acceptance, there are still few minor points that needs to be addressed regarding some typos or English grammar errors. We would like to invite you to carefully reread the text and correct any errors.

Reply: Thanks. We have revised the artical carefully and corrected the typos or English grammar errors. Please refer to the revised manuscript.

Reviewers' comments:

Reviewer's Responses to Questions

Comments to the Author

1. If the authors have adequately addressed your comments raised in a previous round of review and you feel that this manuscript is now acceptable for publication, you may indicate that here to bypass the “Comments to the Author” section, enter your conflict of interest statement in the “Confidential to Editor” section, and submit your "Accept" recommendation.

Reviewer #1: All comments have been addressed

Reviewer #2: All comments have been addressed

2. Is the manuscript technically sound, and do the data support the conclusions?

Reviewer #1: Yes

Reviewer #2: Yes

3. Has the statistical analysis been performed appropriately and rigorously?

Reviewer #1: Yes

Reviewer #2: N/A

4. Have the authors made all data underlying the findings in their manuscript fully available?

Reviewer #1: Yes

Reviewer #2: Yes

5. Is the manuscript presented in an intelligible fashion and written in standard English?

Reviewer #1: Yes

Reviewer #2: Yes

6. Review Comments to the Author

Reviewer #1: The authors answered the asked questions and accordingly modified the manuscript.

However, the text still contains some typos or English grammar errors.

Authors should carefully reread the text and correct any errors.

Reviewer #2: (No Response)

Reply: Thank you for your attention to spelling and grammatical errors in the article. We have revised the artical carefully and corrected the typos or English grammar errors. Please refer to the revised manuscript.

7. PLOS authors have the option to publish the peer review history of their article (what does this mean?). If published, this will include your full peer review and any attached files.

Do you want your identity to be public for this peer review? For information about this choice, including consent withdrawal, please see our Privacy Policy.

Reviewer #1: No

Reviewer #2: No

Reply: Thanks. The uploaded pictures have been adjusted by PACE.

---

## [Editor Report · Decision Letter 2]

24 Feb 2022

The mitochondrial proteomic changes of rat hippocampus induced by 28-day simulated microgravity

PONE-D-21-31461R2

Dear Dr. Ji,

We’re pleased to inform you that your manuscript has been judged scientifically suitable for publication and will be formally accepted for publication once it meets all outstanding technical requirements.

Kind regards,

Angela Chambery, PhD

Academic Editor

PLOS ONE
---

## [Editor Report · Acceptance letter]

1 Mar 2022

PONE-D-21-31461R2 

The mitochondrial proteomic changes of rat hippocampus induced by 28-day simulated microgravity 

Dear Dr. Ji:

I'm pleased to inform you that your manuscript has been deemed suitable for publication in PLOS ONE. Congratulations! Your manuscript is now with our production department. 

Kind regards, 

on behalf of

Dr. Angela Chambery 

Academic Editor

PLOS ONE